# Learning soft interventions in complex equilibrium systems

**Michel Besserve**[1]                    **Bernhard Schölkopf**[1]

[1]Department of Empirical Inference, Max Planck Institute for Intelligent Systems, Tübingen, Germany.

## Abstract

Complex systems often contain feedback loops that can be described as cyclic causal models. Intervening in such systems may lead to counterintuitive effects, which cannot be inferred directly from the graph structure. After establishing a framework for differentiable soft interventions based on Lie groups, we take advantage of modern automatic differentiation techniques and their application to implicit functions in order to optimize interventions in cyclic causal models. We illustrate the use of this framework by investigating scenarios of transition to sustainable economies.

## 1 INTRODUCTION

Designing optimal interventions in complex systems, composed of many interacting parts, is a key objective in multiple fields. In the context of socio-economic systems, the design of public policies to improve economic and social welfare is a major source of scientific and political debate. Moreover, the positive aspects of socio-economic activities need to be traded-off with their environmental impacts, as their long term consequences may considerably affect societies [Dearing et al., 2014, Sherwood and Huber, 2010]. Interestingly, a priori intuitive interventions in such systems may lead to paradoxical outcomes. The rebound effect in energy economy, first reported by Jevons [1866], is paradigmatic: while the energy efficiency of devices may considerably increase due to technological improvements, this may trigger an overall increase of energy consumption due to increases in demand [Brockway et al., 2021]. This suggests in particular that efficiency alone may not be the best way to foster a transition towards sustainability, and calls for a quantitative study of optimal interventions in such complex systems [Arrobbio and Padovan, 2018]. As argued for the case of rebound effects [Wallenborn, 2018], such

unexpected behaviors may reflect balanced causal relationships designed by evolution [Andersen, 2013], and feedback loops [Blom and Mooij, 2021] that maintain a system at an optimal "equilibrium" operating point independent from external perturbations, challenging classical causal inference assumptions of faithfulness and acyclicity.

While interventions have been extensively investigated theoretically in the field of causality [Pearl, 2000, Imbens and Rubin, 2015], the case of systems incorporating feedback loops remains particularly challenging, and therefore led to only limited applications to real-life complex systems. A possible first step to study such systems is to approximate them by a model that operates at an equilibrium point, and can thus be described by a cyclic structural causal model [Bongers et al., 2016]. Such models satisfy a self consistent set of equations that, under unique solvability assumptions, fully identifies the operating point, and allows to study interventions. For practical and ethical reasons, interventions that do not change the causal structure, called soft interventions, arguably provide a more realistic account of changes that can be performed in real life systems. While a restricted set of qualitative results exist for such interventions [Blom et al., 2020], their quantitative assessment and design in complex systems is made difficult by the analytical intractability of the self-consistency relations.

In this paper, we propose a framework for a general class of differentiable parametric soft interventions based on Lie groups and leverage recent technical and algorithmic developments allowing learning implicit functional relationships [Bai et al., 2019] to optimize such interventions. After defining Lie interventions and assaying their theoretical properties, we provide a computational framework to optimize them. We illustrate its application to economic models derived from real data, offering a novel approach to computational sustainability. Proofs are provided in Appendix B. Code is available at https://github.com/mbesserve/lie-inter.

**Related work.**    Various types of economic equilibrium models (EEM) have been used to investigate macroeco-

*Accepted for the 38[th] Conference on Uncertainty in Artificial Intelligence  (UAI 2022).*

nomic effect of specific interventions [Wiebe et al., 2018, Wood et al., 2018]. Also, experimental design in two-sided marketplaces has been investigated in [Johari et al., 2022]. In contrast to such work, we develop a general optimization framework that allows the optimal design of interventions to achieve specific goals. A restricted set of EEMs have been investigated more extensively from an optimization perspective (see, e.g., Esteban-Bravo 2004); however, these are restricted to rather specific assumptions and constraints that allow to address optimization with linear programming approaches. Instead, we rely on automated differentiation and backpropagation algorithms that allow studying mechanisms and interventions with a broad range of non-linearities. In the field of causality, several studies investigate the relationship between the equilibrium of dynamical systems and structural causal models (SCM) [Mooij et al., 2013, Peters et al., 2020] and how the causal structure can be learnt from data. In contrast, we focus on designing soft interventions in an known SCM at equilibrium. While the specificity of soft interventions have started to be investigated theoretically in structural causal models [Rothenhäusler et al., 2015, Kocaoglu et al., 2019, Jaber et al., 2020, Correa and Bareinboim, 2020, Blom et al., 2020], the present work is to the best of our knowledge the first to investigate theoretically and algorithmically the design of soft interventions in cyclic causal models. The algorithmic approach relies on modeling economic equilibrium with deep equilibrium models [Bai et al., 2019]. This approach belongs to the category of implicit deep learning [El Ghaoui et al., 2021], which has been used in a variety of applications such as model predictive control [Amos et al., 2018] and multi-agent trajectory modelling [Geiger and Straehle, 2020].

## 2 MOTIVATION AND BACKGROUND

### 2.1 ENVIRONMENTAL ECONOMIC MODELS

In the face of the increasing severity of climate change and further environmental impacts of human activities, our societies face challenges to transition to more sustainable economies. An overarching difficulty is the complexity of the systems that need to be intervened on, which comprise tightly intertwined components, ranging from economic agents to a broad variety of ecosystems [Haberl et al., 2019].

A classical way to represent the economy and its impacts are input-output (IO) multi-sector economic equilibrium models [Stadler et al., 2018], in which economic activities are divided in $d$ interdependent *sectors* and described by a positive $d$-dimensional *output* vector $x$ (see Appendix A). We take as a guiding example the demand-driven model introduced by Leontief [1951], which is the basis of the *Input-Output analysis* approach to environmental impact assessment. In such models, the sectors' outputs at economic equilibrium $x^*$ are dependent on the vector $y$ gathering final

demand for each product (consumed by users instead of being used to make another product). Satisfying the demand of all sectors implies the self-consistent equation

$$x^* = Ax^* + y, \tag{1}$$

where $A$ is the so-called *technical coefficients matrix*, with $A_{ij}$ the amount of each product $i$ used as input to produce product $j$. An example of technical coefficient matrix estimated from economic data is provided in Fig. 1a. While such equilibria can be thought of as the asymptotic value of $x$ in a dynamic model (see Appendix A) we focus our analysis on the equilibrium equations without consideration for the dynamics that gives rise to it. In turn, the socio-economic impacts (e.g., employment) and environmental stressors (e.g., GHG emissions, water use, ...) of each sector's activity is gathered in a vector of *impacts* $s$ such that

$$s = Rx \tag{2}$$

where $R$ is a *footprint intensity* matrix such that $R_{ij}$ is the amount of impact of type $i$ generated by unit of output $j$. To mitigate major long term negative consequences of environmental stressors (see, e.g., Dearing et al. 2014, Sherwood and Huber 2010), a reorganization of the global economy is required, which may consist in intervening on economic sectors, their impacts and their interactions reflected in the matrices $A$ and $R$. However, this faces three challenges.

***Challenge 1*: social acceptability.** Reducing a sector's activity may lead to both positive environmental effects (yielding lower footprints) and negative socio-economic impacts (such as reducing economic growth and employment, see Appendix E). Decision makers thus trade off environmental goals with the social acceptability of the chosen policies.

***Challenge 2*: recurrence between sectors.** The sectors' activities are tightly intertwined by their reciprocal demands, as illustrated at the bottom of Fig. 1a: electricity production through renewable energy requires wind turbines, which require metals, while the metal industry requires itself electricity to extract metals from ores and transform them. Such cycles make it challenging to anticipate the system-wide consequences of interventions a particular sector.

***Challenge 3*: rebound effects.** The complexity of the economic system also manifests itself through balancing mechanisms that reflect the utility maximization behavior of economic agents, such as rebound effects. Consider $x^*$ in eq. 1, which can be written as a function of final demand

$$x^* = (I - A)^{-1}y. \tag{3}$$

In practice, final demand is influenced by prices of each good and often modeled by a static demand curve $d_i$ for good $i$ such that $y_i = d_i(p_i)$. A final demand rebound through prices can be simulated in the Leontief model as

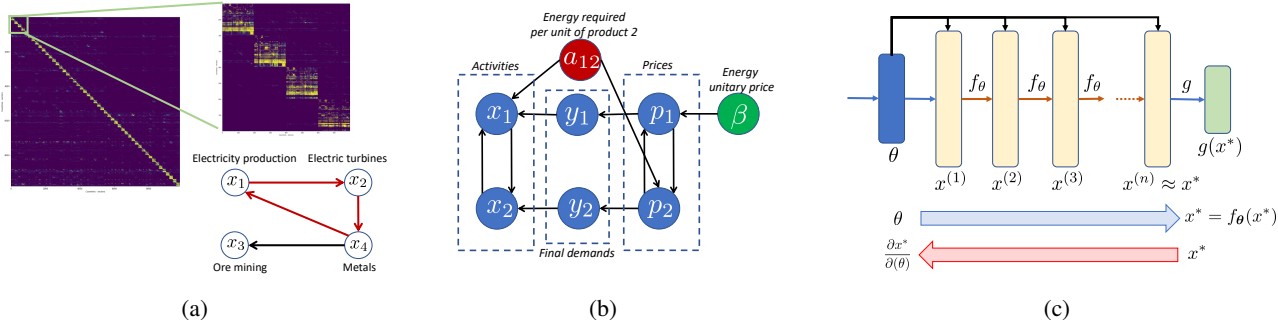

Figure 1: (a) Top left: technical coefficient matrix between 200 sectors and 49 world regions for 2011 (source: Exiobase 3, Stadler et al. 2018). Top right: magnification of the top left corner of this matrix. Diagonal blocks reflect the stronger sector dependency within a country. Bottom right: putative example of cyclic dependency between different sectors. (b) Illustration of the causal graph for *rebound trough prices* in a two sector economy. (c) Principle of deep equilibrium models.

follows. Energy efficiency of the production of a particular good $j$ corresponds to a decrease of $A_{ej}$, where $e$ indicates the energy sector, but this modification also affects the unit price through energy costs. For simplicity, we define the price vector $\boldsymbol{p}$ of goods such that it is proportional to the energy required in all sectors involved in the production of one unit of this good. It can thus be modelled by a self-consistent relation involving the technical coefficient matrix:

$$\boldsymbol{p}^* = A^\top \boldsymbol{p}^* + \beta \boldsymbol{\delta}_e \,,$$

where $\delta_e$ is a canonical basis vector which takes value 1 for the energy sector, and value 0 for all other sectors. For illustrative purposes, the overall causal model is shown in Fig. 1b in the case of a two sector economy, with sector 1 being the energy sector. The price-based rebound mechanisms then operates as follows: a decrease of $A_{ej}$ will decrease energy demand on sector $e$, but will also decrease the unit price of goods for sector $j$ (and downstream sectors consuming its goods). Because the demand curves $d_j$ are monotonically decreasing, the price drop increases the final demand for these products, which in turn increases economic activity according to eq. 3, and their environmental footprint. The rebound may thus be avoided by simultaneously intervening on the unit price of energy $\beta$ through a tax policy, so that price level is maintained high and prevents increases of final demand (see Fig. 1b). Importantly, while eq. (3) provides a linear relationship between activity and final demand, once we assume $\boldsymbol{p}$ is price dependent, the system of equations becomes non-linear and finding an analytic expression of the economic equilibrium is nontrivial. Our approach to designing interventions in cyclic causal models will be applied to models illustrating the above three challenges.

## 2.2 CYCLIC CAUSAL MODELS

Interventions and their effects on systems have been investigated using Structural Causal Models (SCM) [Pearl, 2000]. In this framework, relationships between observed variables

$X_k$ are described by a set of structural assignments

$$X_k := f_k(\mathbf{Pa}_k, \epsilon_k) \,,$$

where $\mathbf{Pa}_k$ indicates the parents of variable $X_k$ in an associated directed causal graph, such as the one illustrated in Fig. 1a. Interventions turn an SCM into a different one, by applying a modification to at least one of its elements. Broadly construed, interventions range from "hard" interventions that modify the structure of the graph to "soft" interventions that do not [Eberhardt and Scheines, 2007].

While in acyclic graphs, interventions have generic effects on their descendants in the causal graph, and no effects on the parents, Blom et al. [2020] have shown that causal effects are less easy to read in graphs containing cycles. Anticipating the effect of interventions in cyclic graphs overall requires to estimate the changes in the equilibrium point, which is typically non-trivial. While a variety of approaches may be used (e.g., based on root finding approaches), designing optimal interventions for self-consistent equations that cannot be handled analytically is challenging, especially in high dimensional systems. Recent work in deep neural network has come up with techniques allowing gradient descent based optimization of such equilibrium models [Bai et al., 2019].

## 2.3 DEEP EQUILIBRIUM MODELS

Bai et al. [2019] introduced deep learning architecture elements with input-output functional relationships $\boldsymbol{x}^* = \mathbf{g}(\boldsymbol{\theta})$ between variables $\boldsymbol{x}^*$ and parameters $\boldsymbol{\theta}$ that are only defined through a self-consistent equation

$$\boldsymbol{x}^* = \mathbf{f}_{\boldsymbol{\theta}}(\boldsymbol{x}^*).$$

Assuming that for each value $\boldsymbol{\theta}$ there is a unique solution $\boldsymbol{x}^*$, the gradient with respect to one parameter component $\theta_k$ can be obtained through another self-consistent equation

$$\frac{\partial \boldsymbol{x}^*}{\partial \theta_k} = \frac{\partial \mathbf{f}_{\boldsymbol{\theta}}}{\partial \theta_k}(\boldsymbol{x}^*) + \frac{\partial \mathbf{f}_{\boldsymbol{\theta}}}{\partial \boldsymbol{x}} \frac{\partial \boldsymbol{x}^*}{\partial \theta_k} \,.$$

Overall, **g** can be integrated as a layer in more complex differentiable models, which, as depicted in Fig. 1c, can be understood as a cascade of multiple layers with identical functions and shared parameters, with specific accelerated fixed point iteration approaches to compute the forward and backward passes [Bai et al., 2019]. In this paper, we use Anderson's acceleration [Walker and Ni, 2011], which essentially generalizes the forward iteration approach (i.e. iterating $\boldsymbol{x}_{k+1} = \mathbf{f}(\boldsymbol{x}_k)$ until convergence) by leveraging the $m$ previous estimates in order to find a better estimate. Overall, these layers offer a differentiable framework for investigating the behavior of cyclic graphs that we use to design interventions.

## 2.4 LIE GROUPS

Using deep equilibrium models, we can learn differentiable soft interventions compatible with classical optimization frameworks. We will use the concept of Lie groups, which are smooth manifolds of transformations (see Appendix A for more background), in order to implement smooth soft interventions. In short, a group $G$ is a set of objects equipped with a group "multiplication" operation mapping $(g_1, g_2) \in G^2$ to $g_1 g_2 \in G$ and an inverse operation $g^{-1}$ with the following properties:

- (associativity) $(g_1 g_2) g_3 = g_1 (g_2 g_3)$,
- (identity element) there exist a unique identity element $e$ such that for all $g$, $eg = ge = g$,
- (inverse) for all $g \in G$, there exists a unique element $g^{-1}$ such that $g g^{-1} = g^{-1} g = e$.

Groups perform transformations on objects in a set $\mathcal{X}$ through the definition of a group action operation $\varphi$ mapping $(g, x) \in G \times \mathcal{X}$ to $\varphi_g(x) = g \cdot x \in \mathcal{X}$, such that

- (identity) for all $x \in \mathcal{X}$, $e \cdot x = x$,
- (compatibility) for all $(g, h) \in G \times G$, for all $x \in \mathcal{X}$, $g \cdot (h \cdot x) = (gh) \cdot x$.

A real Lie group is a group that is also a finite-dimensional real smooth manifold (see Appendix A), in which the group operations of multiplication and inversion are smooth maps. The differentiability of Lie groups will be leveraged to design smooth interventions.

## 3 INTERVENING IN SMOOTH MODELS

### 3.1 SMOOTH CAUSAL GRAPHICAL MODELS

We define a smooth structural causal model (SSCM) as a set of variables $\{x_j\}$ related to each other through structural equations and vertices in a directed graph as follows.

**Definition 1** (SSCM). A $d$-dimensional smooth structural causal model is a 4-tuple $(\mathcal{X}, \mathcal{T}, \mathbb{S}, \mathcal{G})$ consisting of

- two collections of smooth manifolds $\mathcal{X} = \{\mathcal{X}_i\}_{i=1..d}$ and $\mathcal{T} = \{\mathcal{T}_j\}_{j=1..d}$,
- a directed graph $\mathcal{G} = (V, E)$ with set $V$ of $d$ vertices and set $E$ of directed edges between them, each vertex being associated to one variable $x_i \in \mathcal{X}_j$,
- a set $\mathbb{S}$ of structural assignments $\{x_j := f_j(\mathbf{Pa}_j, \theta_j), \theta_j \in \mathcal{T}_j\}_{j=1,...,d}$, where $f_k$ are smooth maps, and $\mathbf{Pa}_j$ are the variables indexed by the set of parents of vertex $j$ in $\mathcal{G}$.

Compared to classical definitions of SCMs (see, e.g., Peters et al. 2017), we have replaced exogenous random variables by deterministic parameters living on a manifold. This general definition does not prevent assigning random variables to some (components of) these parameters. In the cases considered here, $\mathcal{T}_i$ are subsets of Euclidean spaces. We are particularly interested in cyclic SCMs, where there exists at least one directed path linking one vertex to itself. As a consequence, the possible values achieved by each variable have to be chosen among the solutions of the $d$ self-consistent structural equation constraints. We assume the unintervened causal model is locally uniquely solvable.

**Definition 2.** A SSCM is locally uniquely solvable around a reference point $(\boldsymbol{x}^{\text{ref}}, \boldsymbol{\theta}^{\text{ref}})$ whenever there exists a neighborhood $U_{\boldsymbol{\theta}}$ of $\boldsymbol{\theta}^{\text{ref}}$ and a neighborhood $U_{\boldsymbol{x}}$ of $\boldsymbol{x}^{\text{ref}}$ such that for all $\boldsymbol{\theta} \in U_{\boldsymbol{\theta}}$ there exists a unique (self-consistent) solution to the set of structural assignments $\boldsymbol{x}^*(\boldsymbol{\theta}) \in U_{\boldsymbol{x}}$.

Note that this is adapted to our SSCM definition and differs from the unique solvability definition of Bongers et al. [2016], which was formulated for causal models with random exogenous variables. This property is guaranteed by a condition on the Jacobian of the structural equations.

**Proposition 1.** We say the SSCM is locally diffeomorphic at $(\boldsymbol{x}^{\text{ref}}, \boldsymbol{\theta}^{\text{ref}})$ when $(\boldsymbol{x}^{\text{ref}}, \boldsymbol{\theta}^{\text{ref}})$ is a solution and the Jacobian of the mapping $\boldsymbol{x} \to \boldsymbol{x} - \mathbf{f}(\boldsymbol{x}, \boldsymbol{\theta}^{\text{ref}})$ is invertible. Such a SSCM is uniquely solvable around this reference point and the local mapping $\boldsymbol{\theta} \mapsto \boldsymbol{x}^*(\boldsymbol{\theta})$ is smooth.

In the context of IO analysis presented in Section 2.1, the variables can be the sector's outputs and unit prices. For eq. (1), the resulting SSCM thus contains the affine structural assignments associated to each component of $\boldsymbol{x}$

$$\mathbb{S} = \{x_k := \sum_j A_{kj} x_j + y_k\},$$

which are clearly smooth, and the $\{A_{kj}, y_k\}$'s may be assumed fixed or free parameters within an interval.

### 3.2 LIE INTERVENTIONS

We will consider interventions parameterized by an element $u$ that turns the unintervened equilibrium solution

$\boldsymbol{x}^*(\boldsymbol{\theta})$ into the *intervened equilibrium solution* $\boldsymbol{x}^{(u)}(\boldsymbol{\theta})$ over a range of values of $\boldsymbol{\theta}$. In particular, we define Lie interventions implemented through the action of a Lie group.

**Definition 3** (Lie intervention). A Lie intervention on an SSCM $\mathcal{M} = (\mathcal{X}, \mathcal{T}, \mathbb{S}, \mathcal{G})$ with a set of smooth structural assignments $\mathbb{S}$ is a pair $(L, \varphi)$ where $L$ is a Lie group and a smooth group action $\varphi : L \times \mathbb{S} \to \mathbb{S}$. The action defines a family of intervened SSCMs $\mathcal{M}^{(g)} = (\mathcal{X}, \mathcal{T}, \varphi(g, \mathbb{S}), \mathcal{G})$, for $g$ in a neighborhood of the identity within $L$.

Note in particular that applying the identity element of the group leads to the original (unintervened) causal model. Such interventions preserve unique solvability.

**Proposition 2** (Solvability). For a Lie intervention on a locally diffeomorphic SSCM, there is a neighborhood $U_L$ of the identity $e$ in $L$ such that the intervention is soft, the family of intervened SSCMs is locally uniquely solvable and the local mapping to the intervened solution $(g, \boldsymbol{\theta}) \mapsto \boldsymbol{x}^{(g)}(\boldsymbol{\theta})$ is smooth.

**Multiplicative Lie interventions.** A simple way of intervening on an arbitrary system is to multiply one selected assignment by a strictly positive scalar coefficient. We can consider $\mathbb{R}_+^*$ equipped with multiplication as a Lie group, that acts on a node by rescaling its structural assignment. Several such *scalar* Lie interventions can then be combined into a *distributed* intervention on a set of nodes instead of a single one. A group element is then a strictly positive vector $\boldsymbol{\alpha} > 0$ acting on assignments indexed by $I$ such that

$$\boldsymbol{\alpha} \cdot \mathbb{S}_{|I} = \{x_k := \alpha_k f_k(x_k, \theta_k), k \in I\}.$$

In the context of Input-Output models presented in Section 2.1, applying this intervention can be seen as reducing or increasing the demand for products of specific sectors. Reducing the demand for a sector with large GHG emissions is for example a relevant objective for the transition to a sustainable economy and may be implemented by public policy in various ways (taxes, norms, ...). Such interventions are investigated in industrial ecology [Wood et al., 2018].

In the context of our guiding example, the influence of multiplicative interventions has an intuitive real world interpretation. However, *shift interventions* (using the additive group, acting by addition on a structural assignment) may also be an easily interpretable choice in some settings, and have been exploited for causal inference [Rothenhäusler et al., 2015]. Moreover, some settings may require other classical, possibly multidimensional, Lie groups (e.g. Besserve et al. [2018] exploit the group of rotations of the $n$-dimensional Euclidean space $SO(n)$). Finally, in contexts where the model stems from a mechanistic model, e.g. relying on physics equations, Lie interventions that change meaningful model parameters may act on structural equations in more complex ways.

## 3.3 INVARIANT SOFT INTERVENTIONS

The rebound effect is paradigmatic of interventions that may trigger undesired effects that we wish to prevent. To this end, simultaneous interventions on other parts of a system have been considered in applications. For example, a rebound through prices can be prevented by a simultaneous auxiliary intervention of prices through taxes, such that the prices remain invariant to the overall intervention. Using the SSCM framework, we theoretically investigate the conditions under which some variables of the causal model can be maintained invariant to the Lie intervention on others.

**Motivating example.** Consider the following SSCM with parameters $\boldsymbol{\theta} = (\tau, \alpha, \beta, \gamma)$ with distributed multiplicative Lie intervention $\boldsymbol{u}$:

$$\begin{cases} x &= \tau, \\ y &= u_y(\alpha x + \beta z), \\ z &= u_z \gamma y. \end{cases} \quad (4)$$

By choosing $u_z = \frac{1}{u_y}$, the intervened equilibrium solution component $z^{(u)}$ becomes insensitive to multiplicative interventions $(u_y, u_z)$, such that $z^{(\boldsymbol{u})}(\boldsymbol{\theta}) = z^*(\boldsymbol{\theta})$ for arbitrary values of parameters $\boldsymbol{\theta}$ in a neighborhood of the reference parameter (see Appendix C). This result suggests that the influence of soft interventions ($u_y$ in this example) can be restricted to a subset of nodes, by choosing a second intervention ($u_z$ in this example) on an auxiliary variable. However, it is unclear whether this result still holds when the functional assignment of $z$ becomes non-linear.

To frame this question in a general setting, we introduce a class of soft interventions under invariance constraint.

**Definition 4** (Invariant soft interventions). Given an SSCM with Lie intervention from group $L$ on node $i$. The intervention leaves node $j$ *invariant* by *leveraging* node $k$ if for all group elements $u$ in a neighborhood $\mathcal{N}$ of the identity, there exists a soft intervention on node $k$, $f_k^{(u)}(\mathbf{Pa}_k, \boldsymbol{\theta})$, replacing functional assignment $f_k$ such that the intervened node value $x_j^{(u)}$ satisfies $x_j^{(u)}(\boldsymbol{\theta}) = x_j^*(\boldsymbol{\theta})$ in a neighborhood of the reference parameter. Node $i$ is called the intervened node, node $j$ is called the invariant node, and node $k$ is called the auxiliary node.

**Remarks:** The soft intervention property is key, as it entails that the use of an auxiliary variable to enforce the invariance constraint must only exploit the information available to this node as defined by its parents in the unintervened graph (and no parameter values). This constraint makes deployment more realistic in a complex system, as intervening does not require supervision by an external entity monitoring the whole system. Unless otherwise stated, the auxiliary node will be chosen identical to the invariant node.

Let us denote $\boldsymbol{x}_{-j}$ and $\mathbf{f}_{-j}$ the vector and mapping with the $j$-th component removed. We also define two quantities

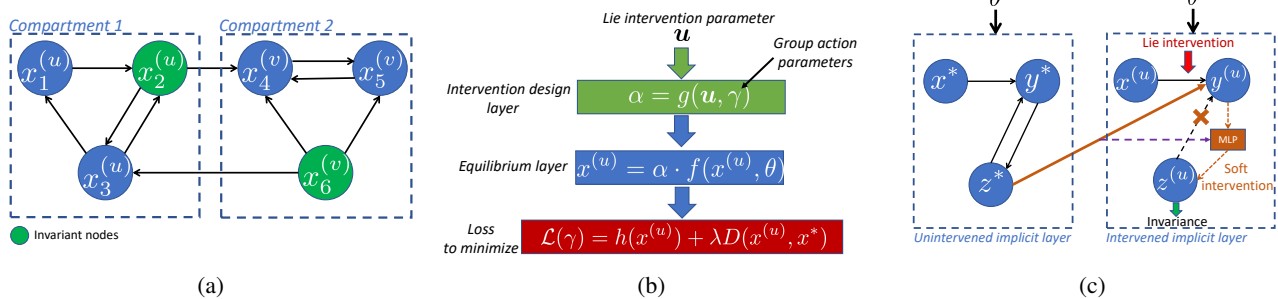

Figure 2: (a) Illustration of a compartmentalized intervention: enforcing invariance of the green nodes allows each compartment to be independently influenced by two (invariant) interventions $u$ and $v$. (b) Architecture for Lie intervention optimization. The equilibrium layer is controlled by intervention parameters and a loss is applied to its output. (c) Schematic representation of the procedure to learn invariant soft intervention ($y$: intervened node, $z$: invariant and auxiliary node). A multilayer perceptron (MLP) learns the soft intervention enforcing invariance of $z^{(u)}$ over a range of parameter values.

important for the existence of such interventions. The partial derivative $\frac{\partial x_j^*}{\partial x_k}\big|_{\boldsymbol{\theta}=\boldsymbol{\theta}^{\text{ref}}}$ is obtained by performing a hard intervention $x_k = \lambda$ leading to equilibrium value $x_j^{(\lambda)}(\boldsymbol{\theta}^{\text{ref}})$, and computing the derivative $\frac{dx_j^{(\lambda)}}{d\lambda}\big|_{\lambda=x_k^*(\boldsymbol{\theta}^{\text{ref}})}$. The Jacobian $J_{x_{\mathbf{Pa}_k}^*}^{\boldsymbol{\theta}}(\boldsymbol{\theta}^{\text{ref}})$ is the Jacobian of the mapping from the parameters $\boldsymbol{\theta}$ to the vector consisting of the parent nodes of $k$ at equilibrium. Based on these two quantities, we have the following sufficient condition.

**Proposition 3.** Consider an SSCM locally diffeomorphic at $(\boldsymbol{x}^{\text{ref}}, \boldsymbol{\theta}^{\text{ref}})$ with intervened/invariant/auxiliary triplet of nodes $(i, j \neq i, k \neq i)$. If the Jacobian of the mapping $\boldsymbol{x}_{-j} \to \boldsymbol{x}_{-j} - \mathbf{f}_{-j}(\boldsymbol{x}_{-j}, \boldsymbol{\theta}^{\text{ref}})$ is invertible, $J_{x_{\mathbf{Pa}_k}^*}^{\boldsymbol{\theta}}(\boldsymbol{\theta}^{\text{ref}})$ has full column rank, and $\frac{\partial x_j^*}{\partial x_k}\big|_{\boldsymbol{\theta}=\boldsymbol{\theta}^{\text{ref}}} \neq 0$, then the intervention on $i$ leaves node $j$ invariant by leveraging node $k$.

This result suggests that the motivating example of eq. (4) can be extended, in a neighborhood of the identity, beyond the linear case, when the number of free parameters considered remains low relative to the number of parents of the auxiliary node. However, as can be seen in the proof, the soft intervention on the auxiliary variable is given by an implicit function theorem, suggesting non-parametric models are necessary to learn it (based e.g. on automated differentiation methods). This will be described in Sec. 4.

### 3.4 COMPARTMENTALIZED INTERVENTIONS

Invariant interventions allow to restrict the propagation of effects to a subset of nodes. If a complex system can be partitioned into sparsely connected subsets of nodes, we can consider designing such interventions in order to modify the equilibrium values of each compartment independently from each other.

**Definition 5.** Given a partition of the SSCM nodes into

$K$ compartments $\{C_k\}_{k=1,...,K}$. Given interventions on each compartment, parameterized by respective parameters $u_k$, leading to the intervened SSCM equilibrium solution $\boldsymbol{x}^{(u_1,...,u_k,...,u_K)}(\boldsymbol{\theta})$. Interventions are compartmentalized when for all $k$, for all nodes $j \in C_k$, component $\boldsymbol{x}_j^{(u_1,...,u_k,...,u_K)}(\boldsymbol{\theta})$ does not depend on $u_m$ for $m \neq k$.

The following result guarantees that if the nodes influencing other compartments are made invariant, interventions on each compartment can be designed and performed independently from each other as their effects remain confined to their own compartment.

**Proposition 4.** Given a partition $\{C_k\}$ of the SSCM nodes. If for each compartment $k$ there exists one invariant soft intervention performed on structural equations such that intervened, auxiliary and invariant nodes belong to the compartment, and all nodes of this compartment that have an outgoing arrow pointing to a different compartment are invariant, then those interventions are compartmentalized.

A fundamental aspect of this result is that, from the definition of invariant interventions, compartmentalization is valid over a range of parameters of the causal model (a neighborhood of the reference point) and a range of Lie interventions parameters (a neighborhood of the identity). This can be seen as a way to enforce interpretability of interventions by restricting their influence to a specific subsystem, at least for a range of parameter values. An illustration of a setting compatible with Prop. 4 is provided in Fig. 2a, where the equilibria of two sparsely connected compartments are interdependent (notably, the causal ordering algorithm described in Blom et al. [2020] returns a single cluster merging both compartments). Enforcing invariance of the green nodes, each associated to one intervention ($u$ and $v$) within their compartment allows applicability of Prop. 4.

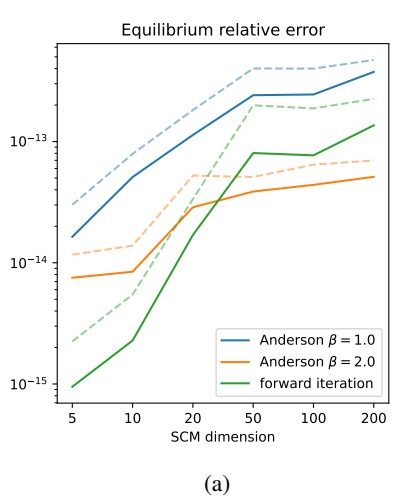
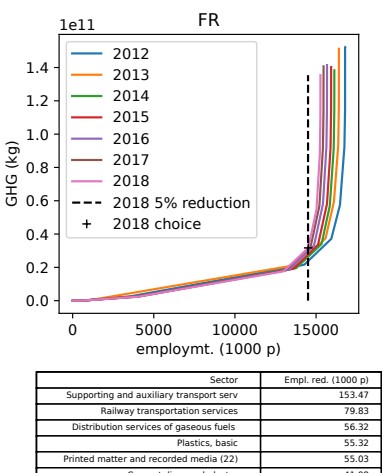
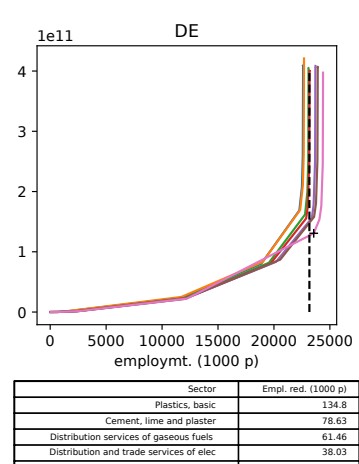

| | Sector | Empl. red. (1000 p) |
|---|---|---|
| | Supporting and auxiliary transport serv | 153.47 |
| | Railway transportation services | 79.83 |
| | Distribution services of gaseous fuels | 56.32 |
| | Plastics, basic | 55.32 |
| | Printed matter and recorded media (22) | 55.03 |
| | Cement, lime and plaster | 41.98 |

| | Sector | Empl. red. (1000 p) |
|---|---|---|
| | Plastics, basic | 134.8 |
| | Cement, lime and plaster | 78.63 |
| | Distribution services of gaseous fuels | 61.46 |
| | Distribution and trade services of elec | 38.03 |
| | Dairy products | 35.39 |
| | Air transport services (62) | 32.67 |

|  (a)  |  (b)  |  (c)  |
|---|---|---|

Figure 3: (a) Equilibrium relative error for different methods and SCM dimensions (solid: mean, dashed: mean+std). (b-c) Outcome of Lie intervention optimization on country models of GHG emission reduction in France (b) and Germany (c) for varying values of $\lambda$ in eq. (5), based on economic models estimated from different years. For year 2018, dashed lines indicates 5% reduction in employment and the cross the corresponding $\lambda$ choice. Tables show sectors with largest employment reduction for 2018.

## 4  INTERVENTION DESIGN

To address *Challenge 2* of Sec. 2.1, we design interventions with implicit layers (see Appendix D for additional details).

**Differentiable architecture.** Base optimization relies on a differentiable architecture comprising one central module representing the cyclic SSCM. Essentially, the cyclic model is represented by an equilibrium layer following Bai et al. [2019], schematized in Fig. 1c: the differentiable module is designed such that forward and backward passes through the equilibrium layer use Anderson acceleration to solve a fixed point equation. This equilibrium layer is cascaded if necessary with parametric layers to achieve specific goals. The architectures are implemented using the PyTorch library.

**Lie intervention optimization** We design an architecture around the equilibrium module to optimize multiplicative intervention according to a loss, as represented in Fig. 2b. Parameters $\boldsymbol{u}$ of the Lie group element are optimized in order to minimize an objective $\mathcal{L}(\boldsymbol{x}^{(u)})$ achieved by the equilibrium solution of the SSCM. This objective may include an additional regularization term, $D(\boldsymbol{x}^{(u)}, \boldsymbol{x}^*)$ with regularization parameter $\lambda$, to enforce that some properties of the intervened system remain invariant or close to the original, non-intervened, equilibrium solution $\boldsymbol{x}^*$.

**Learning invariant interventions.** In order to enforce invariance of interventions based on Sec. 3.3, we follow the procedure exemplified in Fig. 2c. We design two implicit layers with shared parameters $\boldsymbol{\theta}$, the first layer being unintervened giving the corresponding equilibrium values of the

nodes, and the second one being invariantly intervened, for a fixed value of Lie intervention $u$ on the intervened node. In the intervened layer, we replace putative incoming arrows from the invariant node by arrows from the same node in the unintervened graph (as this replacement encodes the invariance assumption) and we replace the functional assignment of the auxiliary node by a Multi-Layer Perceptron (MLP), relying on universal approximation properties to learn a soft intervention that satisfies invariance. We use a least square loss between the intervened and unintervened equilibrium values of the invariant node in order to train the MLP.

## 5  EXPERIMENTS

The following toy and semi-synthetic experiments illustrate how our framework contributes addressing sustainability challenges exposed in Sec. 2.1.

**Evaluation of equilibrium estimation.** We first evaluate the performance of equilibrium layers in computing an accurate estimate of the SSCM solution $\boldsymbol{x}^*$. For that we use the SSCM associated to the economic equilibrium of equation (1) where we select a subset of sectors in order to vary the dimension of $A$. The full matrices $A$, as well as the final demands $\boldsymbol{y}$ are estimated from the Exiobase 3 dataset [Stadler et al., 2018] for years 2012-2018, using the *Pymrio* library [Stadler, 2021] for five countries (France, Germany, Italy, USA, Great-Britain). We compare Anderson acceleration (see [Walker and Ni, 2011]) for two different choices of the mixing parameter $\beta$, together with the baseline forward iteration approach that simply consists

in iterating $\boldsymbol{x}_{k+1} = \mathbf{f}(\boldsymbol{x}_k)$. For each choice of dimension and fixed-point iteration algorithm, we compute the relative error $\frac{\|\boldsymbol{x}^* - \mathbf{f}_{\boldsymbol{\theta}}(\boldsymbol{x}^*)\|}{\|\boldsymbol{x}^*\|}$. The results, averaged across countries and years, show that although forward iteration is the most accurate in lower dimensions, Anderson acceleration with a relaxation parameter $\beta = 2.0$ performs better for SSCM dimensions larger than 50. Interestingly, Anderson acceleration with $\beta = 1.0$ gives the worst performance, suggesting an appropriate choice of $\beta$ is key.

**Optimization of multiplicative Lie interventions.** In order to investigate *Challenge 1*, we optimize the IO demand driven model of eq. (1). The matrices $A$ and $R$, as well as the final demands $\boldsymbol{y}$ and sector output at equilibrium $\boldsymbol{x}^*$ are estimated from yearly activity available in the Exiobase 3 dataset [Stadler et al., 2018], using the *Pymrio* library [Stadler, 2021]. While the data describes economic interactions across multiple countries, we design an economic equilibrium model of each country by neglecting those interactions, and extracting the blocs of matrices $A$ and $R$ relevant to the country under consideration. We design a distributed multiplicative Lie intervention on the activity of all 200 sectors of the database. The coefficient vector $\boldsymbol{\alpha}$ is then optimized in order to reduce the overall greenhouse gas (GHG) emissions cumulated across sectors (estimated by one component of the stressor vector $\boldsymbol{s}$), while enforcing that the overall employment distribution over the sectors stays closest to the non-intervened economy, in order to mitigate challenges associated to reorganizing of economic activities (e.g. mass unemployment and the need for large scale professional reorientation programs). Using the $\ell_1$ norm for regularization, this leads to the following loss:

$$\mathcal{L}(\boldsymbol{u}) = \boldsymbol{c}^\top \boldsymbol{x}^{(u)} + \lambda \|\boldsymbol{e}^{(u)} - \boldsymbol{e}^*\|_1 \qquad (5)$$

where $\boldsymbol{c}$ is the GHG emission intensity of each sector, and $\boldsymbol{e}^{(u)}$ and $\boldsymbol{e}^*$ the intervened and unintervened distributions of employment across sectors (estimated by entry wise multiplication of $\boldsymbol{x}^*$ with one row of matrix $R$). The graphs shown in Figs. 3b-3c (top), illustrate the trade off between employment preservation and GHG emission reduction achieved by varying $\lambda$ for two different countries. Interestingly, the left tail of these curves reflect differences across countries, with Germany having less room than France for reducing emissions before starting reducing employment significantly. The sectors yielding the largest employment reduction also differ across countries, likely influenced both by the overall structure of each economy.

**Control of rebound effects.** To illustrate how *Challenge 3* of Sec. 2.1 can be addressed, we used our invariant intervention framework to prevent price rebound effects. We use a toy 3-sector model, with one energy sector and one target sector for which energy efficiency is increased, modeled by a multiplicative Lie intervention on the energy requirements coefficient of the Leontief matrix. The final demand of this

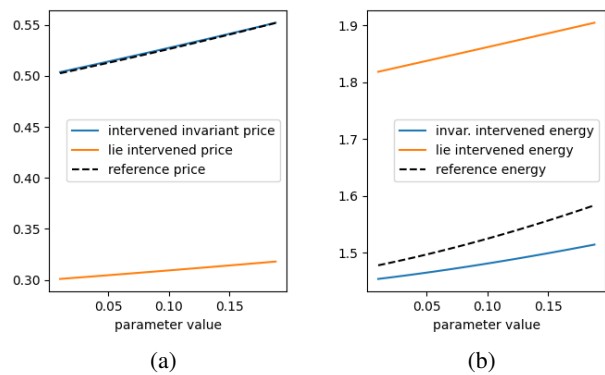

(a) (b)

Figure 4: Outcome of (non-invariant) Lie and invariant interventions on energy efficiency, compared to reference (unintervened) values, in the rebound model described in Fig. 1b: (a) unit price of the target sector, (b) total energy demand.

target sector is taken as invariant node, and controlled by softly intervening on it through a modification of the unit price of this sector. The invariant intervention is learnt using an MLP with two hidden layers (see Appendix D for details). Fig. 1b describes the two quantities that are intervened on: we make a multiplicative Lie intervention on the parameter represented by the red node (energy efficiency), and make sure there is no rebound by making the node invariant to the drop of energy costs using an adaptive taxing policy. Fig. 4a-4b compares 3 models: unintervened (called "reference" in the figure), Lie intervened (without enforcing invariance), and invariantly intervened. For a range of one parameter left free in the Leontief matrix, the results show the invariant intervention maintains the price close to the unintervened model (Fig. 4a), while this price is much lower for the Lie intervention (due to the rebound effect). The benefit of invariance is demonstrated by the effect on the activity of overall energy demand of the economy (Fig. 4b): for the Lie intervention, the rebound through prices leads to the so-called *backfire* scenario: the actual energy savings are negative because usage increased beyond potential savings. In contrast, invariant intervention leads to a reduction of energy demand (relative to the unintervened system), as the rebound through prices is prevented.

**Compartmentalized interventions design.** We further implement compartmentalized interventions and show its benefits for addressing *Challenge 2* in Sec. 2.1 in multisector economic models. We design a two compartment Leontiev model according to Fig. 2a. We optimize two invariant interventions, $u$ on compartment 1 and $v$ on compartment 2, to follow the conditions of Prop. 4. The results provided in Fig. 5, show the invariant node of compartment 1 is unchanged by both values of $u$ and $v$ (Fig. 5a), while the intervened node of this compartment changes value only as a function of its corresponding intervention $u$ (Fig. 5b), in a way similar to the (non-invariant) Lie intervention.

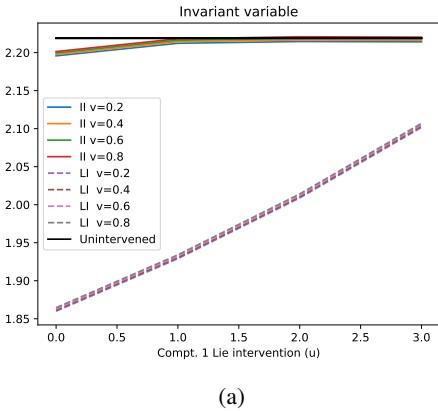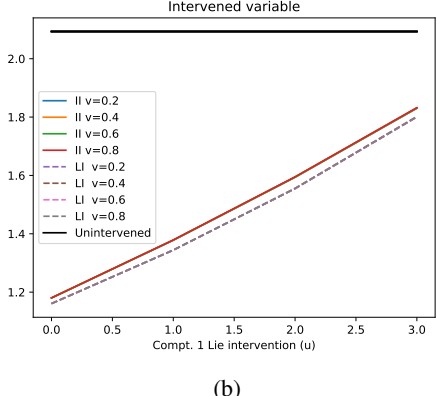

(a)                (b)

Figure 5: (a-b) Outcome of the design of compartmentalized interventions for model of Fig. 2a. The unintervened node values (in black) are compared to invariant interventions (II, solid lines) and their corresponding Lie intervention (LI, dashed lines) (without enforcing invariance), for multiple values of the Lie interventions' parameters $(u, v)$. (a) Value of the invariant node in compartment 1. (b) Value of the intervened node in compartment 1.

# 6 DISCUSSION

We discuss here some limitations of our approach.

**Linearity of the economic model** The linear input-output model that we use should be understood as one way of modeling interactions between economic sectors, commonly used in environmental economics. It was chosen for its interpretability, illustrative purpose, practical relevance, and because there are established approaches to estimate parameters from economic data. However, it should not be understood that economic models are always linear. Note also that we combine this model with a non-linear demand mechanism to study rebound effects (see Challenge 3 and Section 5). Overall, moving towards non-linear models, as allowed by our setting, is in line with the development of computational models in economy, and notably Integrated Assessment Models (IAM) investigating the complex interactions between climate change and societies.

**The case of multiple equilibria.** The equilibrium picked by the equilibrium layer depends on the initialization of the estimate of the equilibrium point in the fixed point iteration algorithm implemented by this layer (this can be described using the notion of "basin of attraction"). While the theory and algorithms developed in this paper focus on the behavior of the causal model in a neighborhood of an given unintervened equilibrium, a prealable grid search for all equilibria may be performed in the most general setting. This may be avoided for the following reasons. From a theoretical perspective, conditions of existence and uniqueness of equilibria are available for many classical models. For example in our application, the Hawkins–Simon condition guarantees the existence of a non-negative output vector that solves the equilibrium relation [Hawkins and Simon, 1949].

From a practical perspective, we are often interested mainly in intervening on the empirically observed equilibrium. For models based on unintervened observed data, we can thus check that the simulated unintervened equilibrium matches the observed data. If however there is a mismatch between the equilibrium obtained by the deep equilibrium layer and the one we are interested in, we can enforce the initialization of the fixed point iteration algorithm in the neighborhood of the expected equilibrium. Our experiments were run with a fixed initialization of the equilibrium point (zero).

# 7 CONCLUSION

We introduced a differentiable soft intervention design framework for general equilibrium systems. We argue those are more likely to approximate deployable interventions in real-world complex systems, e.g. to address key challenges of the transition to sustainable economies. Theoretical results and algorithmic tools are provided to design interventions with desirable invariance properties under the assumption that the considered system is in equilibrium and model parameters are known. Further work in this direction will need to address identifiability of the considered models from observational or experimental data.

**Acknowledgements**

MB is grateful to Philipp Geiger for insightful discussions. This work was supported by the German Federal Ministry of Education and Research (BMBF): Tübingen AI Center, FKZ: 01IS18039B; and by the Machine Learning Cluster of Excellence, EXC number 2064/1 - Project number 390727645.

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
