# OpenReview forum: "Learning soft interventions in complex equilibrium systems"
_auai.org/UAI/2022/Conference — UAI 2022 Oral_

### Official Review · Reviewer_WhzX · 2022-03-31

**Q2(1) Originality/Novelty:** 2
**Q2(2) Significance/Impact:** 2
**Q2(3) Correctness/Technical Quality:** 1
**Q2(6) Clarity Of Writing:** 1
**Q6 Overall Score:** 3
**Q8 Confidence In Your Score:** 2

**Q1 Summary And Contributions:**

The paper proposes a class of smooth structural causal models and a class of interventions for such models. Based on these they then propose a method to identify interventions with certain desired properties.


**Q2 Assessment Of The Paper:**

More detailed information regarding each of these aspects is given below:

**Q2(4) Quality Of Experiments (Optional):**

2: Fair: The experimental evaluation is weak: important baselines are missing, or the results do not adequately support the main claims.

**Q2(5) Reproducibility:**

2: Fair: Key resources (e.g., proofs, code, data) are unavailable but key details (e.g., proof sketches, experimental setup) are sufficiently well-described for an expert to confidently reproduce the main results.

**Q3 Main Strengths:**

The paper does a good job motivating its problem in Section 2.1 and the writing is in general good.


**Q4 Main Weakness:**

The paper is often sloppy in its definitions and formal statements. This is especially problematic as the setting it considers is technically complex making technical precisesness especially important. As a result it is often difficult to understand what exactly is being proposed. The simulation section is also very light on details and therefore does not help make things clearer.

**Q5 Detailed Comments To The Authors:**

General:
- The bold f in Proposition 1 is not formally defined.
- In Proposition 2: what does it mean for an intervention to be "soft". This is not formally defined.
- Definition 4: what is an intervention on a node i. This is not formally defined. Also what is x_j^* in here.
- Paragraph before Prop. 3: What do you mean with "such interventions"? Also the last sentence is very vague and not at all clear to me. What does "parent nodes of k at equilibrium mean".
- Definition 5 is extremely vague.
- The entirety of Section 4 is very vague and this is where it seems the actual method of the paper is being proposed. More mathematical expressions would be helpful.
- In Section 5 I am quite confused what exactly is done with real data and what with synthetic data and how the synthetic data is generated.

Writing:
- The last paragraph of the left column of page 2 has some writing issues
- First paragraph of Section 2.3: why does it say "... pair theta there is a unique solution x*"? Pair of what?
- Prop. 4: what is the compartment k?

**Q7 Justification For Your Score:**

While the paper seems to contain some interesting ideas, these are not presented in a sufficiently rigorous manner and as a result it is hard to evaluate whether they are correct and useful.

**Q9 Complying With Reviewing Instructions:**

1: Yes.

---

### Official Review · Reviewer_bSjC · 2022-04-10

**Q2(1) Originality/Novelty:** 3
**Q2(2) Significance/Impact:** 3
**Q2(3) Correctness/Technical Quality:** 3
**Q2(6) Clarity Of Writing:** 2
**Q6 Overall Score:** 7
**Q8 Confidence In Your Score:** 4

**Q1 Summary And Contributions:**

Authors start their work by a motivation from economics, where there is a need to carefully design interventions in order not to disturb certain parts of the underlying system. They focus on soft-interventions and first establish existence of such desired interventions with mostly non-constructive proofs. They argue this suggests a need for implicit intereventions at which point they propose a neural network based model to search for best interventions.

**Q2 Assessment Of The Paper:**

More detailed information regarding each of these aspects is given below:

**Q2(4) Quality Of Experiments (Optional):**

4: Excellent: The experimental evaluation is comprehensive and the results are compelling.

**Q2(5) Reproducibility:**

2: Fair: Key resources (e.g., proofs, code, data) are unavailable but key details (e.g., proof sketches, experimental setup) are sufficiently well-described for an expert to confidently reproduce the main results.

**Q3 Main Strengths:**

Experiments seem compelling. Motivating the assumptions through existence proofs makes sense.

**Q4 Main Weakness:**

Experimental details could be more elaborate. The presentation could be improved and the literature review could be more exhaustive.

**Q5 Detailed Comments To The Authors:**

Overall, I did enjoy reading the paper and I think it provides a good contribution. Especially the experimental results seem impressive and it might even be better to spend more space on interpreting the results and giving experimental details. However, I have some remarks and questions that would help clarify certain parts of the submission.

Detailed feedback to authors:

"In the literature ... structural causal models (SCM) [Mooij et al., 2013, Peters et al., 2020] and how the causal structure can be learnt from data."
There are some recent papers in this exact literature that solves causal discovery problem from soft-interventional data. The authors should consider adding these relevant citations here:
- Kocaoglu et al. "Characterization and Learning of Causal Graphs with Latent Variables from Soft Interventions," NeurIPS 2019.
- Jaber et al. "Causal Discovery from Soft Interventions with Unknown Targets: Characterization and Learning," NeurIPS 2020.

The presentation could be improved significantly. For example, some paragraphs are way too long to easily follow the argument which is being made.

Authors claim that decision-makers trade off environmental goals with the social acceptability of the chosen policies because positive environmental effects lead to negative socio-economic impacts. I am not sure if this assessment is accurate. Please provide some sources that accept and some that oppose this view.

please define p_i in pg 2 when it appears the first time.

Could you explain the equation at the end of page 2? It is too opaque currently. Where is f(x*,y) coming from and what is (.)?

"for each value pair θ there is a unique solution x∗"
could you also comment on this assumption?

Proposition 1 is great to have as it establishes the validity of the assumption in Definition 2.

Could you make clear what "intervention is soft" claim means? An intervention is presented as a change in the set U but the proof suggests the softness of an extended parameterization.

I do appreciate the authors cross-referencing the discussions in Section 2 throughout their development, e.g., at the end of Section 3.2. But there still seems to be some disconnect in the following sense: It seems that the most widely accepted economic theories make a linearity assumption. However, the rest of the paper and technical contribution lies in learning nonlinear models. Can you comment on this gap and on whether economists would see this method as an acceptable solution?

Proposition 4 talks about "invariance of outgoing arrows" which I believe has not been defined so far. The only notion of invariance so far was invariance of the solution of a node x. What does it mean for an arrow to be invariant? Also, could you comment on why this has to be true in the proof of Proposition 4: "invariance of the outgoing node ensures the equilibrium values of (potentially intervened upon) other compartments"?

Could you comment on how the symmetry properties of a Lie group are enforced in the proposed deep learning architecture? Specifically, why is the suggested intervention a Lie intervention?

Cross-references of some figures are missing. E.g., in the first subsection of Section 5.

Results of Fig 3b,c are very intriguing. How did you pick the employment threshold for different countries?

The last two subsections of Experiments could benefit from describing experiment details a bit more in detail. How was the invariant/Lie intervention determined?

"identifiability of the considered models from observational or experimental data" is left for future work as the authors mention in the Conclusion.

some minor fixes
- equilibrium x∗ are depend on the vector y
- SSCM is called a triplet but it is a 4-tuple.
- familly of intervened SSCMs
- choosing an second
- guaranties
- an extracting the blocs of matrices
- muli-sector

**Q7 Justification For Your Score:**

Paper motivates the problem and the proposed solution well and is complemented with strong experimental results.

**Q9 Complying With Reviewing Instructions:**

1: Yes.

---

### Official Review · Reviewer_hGDj · 2022-04-12

**Q2(1) Originality/Novelty:** 3
**Q2(2) Significance/Impact:** 3
**Q2(3) Correctness/Technical Quality:** 3
**Q2(6) Clarity Of Writing:** 3
**Q6 Overall Score:** 7
**Q8 Confidence In Your Score:** 3

**Q1 Summary And Contributions:**

This paper proposes a new method to design intervention experiments in cyclic graphs. In causal inference, cyclic graphs are usually hard to estimate. However, they exist in many applications. Despite its great impact, existing research for studying cyclic graphs is still limiting. In this paper, the authors provide a new interesting method to learn soft intervention design in cyclic graphs, thereby filling the gap in existing research.

**Q2 Assessment Of The Paper:**

More detailed information regarding each of these aspects is given below:

**Q2(4) Quality Of Experiments (Optional):**

3: Good: The experimental evaluation is adequate, and the results convincingly support the main claims.

**Q2(5) Reproducibility:**

3: Good: Key resources (e.g., proofs, code, data) are available and key details (e.g., proofs, experimental setup) are sufficiently well-described for competent researchers to confidently reproduce the main results.

**Q3 Main Strengths:**

As have mentioned in the summary section, cyclic graphs is an important topic in causal inference. In many applications we may encounter data from cyclic graphs, such as demand and supply data as well as data from biological networks. Despite its significant impact, existing research in cyclic graphs seems still limiting. In this paper, the authors proposed a new approach to design intervention experiments in cyclic system when the system converges to equilibrium. To me, I think it is an important first step for the better study of cyclic graphs in equilibrium systems.

**Q4 Main Weakness:**

The current writing seems a bit unfriendly to the audience that are not very familiar with Lie groups. I think the writings can be further improved for this group of audience. This could be very helpful to expand the impact of this paper. Of course, the authors can skip this comment if they think the current writing is too hard to improve.

**Q5 Detailed Comments To The Authors:**

1. The writing about Lie groups can be further improved.

2. The authors may consider citing the following papers which also consider field experiments in equilibrium systems:

https://pubsonline.informs.org/doi/abs/10.1287/mnsc.2020.3844
https://arxiv.org/abs/2002.05670

**Q7 Justification For Your Score:**

Please see my summary of strength and weakness.

**Q9 Complying With Reviewing Instructions:**

1: Yes.

---

### Official Review · Reviewer_SUBs · 2022-04-13

**Q2(1) Originality/Novelty:** 3
**Q2(2) Significance/Impact:** 3
**Q2(3) Correctness/Technical Quality:** 3
**Q2(6) Clarity Of Writing:** 3
**Q6 Overall Score:** 7
**Q8 Confidence In Your Score:** 3

**Q1 Summary And Contributions:**


The authors investigate soft interventions in cyclic causal models.
By a soft intervention the values of variables are changed, but the structure stays the same. They show how to perform interventions on some variables while keeping other variables unchanged/invariant.
They give a real-life motivation to intervene on the economy to reduce green house gases, and perform simulations.

**Q2 Assessment Of The Paper:**

More detailed information regarding each of these aspects is given below:

**Q2(4) Quality Of Experiments (Optional):**

3: Good: The experimental evaluation is adequate, and the results convincingly support the main claims.

**Q2(5) Reproducibility:**

3: Good: Key resources (e.g., proofs, code, data) are available and key details (e.g., proofs, experimental setup) are sufficiently well-described for competent researchers to confidently reproduce the main results.

**Q3 Main Strengths:**


They prove a new theory of soft interventions  (never mentioned in literature before afaik)

They have a real-life motivating example

They have experiments

**Q4 Main Weakness:**

Parts of the paper are hard too understand

**Q5 Detailed Comments To The Authors:**


What happens when there are multiple different equilibria? Does the method just pick one at random?

Does it really need Lie groups? What happens if the interventions are group that is not a Lie group?


Although the Lie group definition is very general, you only use multiplicative interventions in the rest of the paper?


>p3 x∗ = fθ (x∗) ... Overall, g can be integrated

is f and g the same function?

>p4 We assume the base causal model is locally uniquely solvable.

What is a _base_ causal model?

>Figure 4

It is hard to see the different lines. Especially on (d) I only see three lines.

>p8 3 models: unintervened (called “ref-erence” in the figure), Lie intervened (without enforcing invariance), and invariantly intervened.

The invariantly intervened one is also a Lie intervention, is it not? Then better call them non-invariant and invariant intervened.

I am not sure I fully understand what happens in these experiments

Typos:

>p6 guaranties

>p7 gaz


**Q7 Justification For Your Score:**

I did not really find a weakness

**Q9 Complying With Reviewing Instructions:**

1: Yes.

---

### Decision · Program_Chairs · 2022-05-15

**Decision:**

Accept (Oral)

**Comment:**

Meta Review: **Summary:**  Three out of four reviewers are quite enthusiastic about the paper with clear ‘accepts’, albeit at moderate confidence as the paper was generally considered quite challenging, primarily because of the technical and conceptual novelty of the proposed approach. One reviewer advocated ‘reject’, but with low confidence. On closer inspection most of the complaints by the latter relate to minor/secondary issues, likely stemming from an unfamiliarity with the subject, that should be easily resolved in a final version of the paper, and therefore I will take the other three reviews as leading.

Nevertheless, it is clear the approach presented (using Lie groups to model and optimise differentiable soft interventions in cyclic dynamical systems in equilibrium) is quite demanding, but that is only to be expected given the complexity of the problem. I found the paper very readable for the most part, so I think the authors did as well as can be expected on that front.

On the whole I very much agree with the three reviewers: the paper offers an exciting new approach to experimental design and novel insights into analysis of complex causal systems in equilibrium. The Lie groups are not commonplace in existing causal literature, but represent a natural modelling framework for (soft) interventions in such systems. That means the paper could have significant impact and stimulate new ways to close the gap between standard SCMs and real-world dynamical systems.

In short: clear accept for oral.

**Quality:** High quality, with good results, backed up by experiments and solid theoretical support.

**Clarity:** Good introduction and description. Sometimes in the second half a bit hard to follow on a first reading, but the authors did a good job trying to explain and motivate their approach within the given page restrictions.

**Originality:** Novel approach to modelling soft interventions in cyclic systems. Builds on recent work on deep equilibrium models, and brings the well established mathematical theory of Lie groups into the realm of causal analysis.

**Significance:** Unlikely to be the definitive article itself, but with potential to lead to substantial new developments in the field.